# A 2 μm Wavelength Band Low-Loss Spot Size Converter Based on Trident Structure on the SOI Platform

**DOI:** 10.3390/mi15040530

**Published:** 2024-04-15

**Authors:** Zhutian Wang, Chenxi Xu, Zhiming Shi, Nan Ye, Hairun Guo, Fufei Pang, Yingxiong Song

**Affiliations:** The Key Laboratory of Specialty Fiber Optics and Optical Access Networks, School of Communication and Information Engineering, Shanghai University, Shanghai 200444, China; 1980013841@shu.edu.cn (Z.W.); xu_cx@shu.edu.cn (C.X.); zhiming@shu.edu.cn (Z.S.); hairun.guo@shu.edu.cn (H.G.); ffpang@shu.edu.cn (F.P.); herosf@shu.edu.cn (Y.S.)

**Keywords:** spot size converter, photonics passive devices, integrated photonics, 2 μm

## Abstract

A 2 μm wavelength band spot size converter (SSC) based on a trident structure is proposed, which is coupled to a lensed fiber with a mode field diameter of 5 μm. The cross-section of the first segment of the tapered waveguide structure in the trident structure is designed as a right-angled trapezoidal shape, which can further improve the performance of the SSC. The coupling loss of the SSC is less than 0.9 dB in the wavelength range of 1.95~2.05 μm simulated by FDTD. According to the experimental results, the lowest coupling loss of the SSC is 1.425 dB/facet at 2 μm, which is close to the simulation result. The device is compatible with the CMOS process and can provide a good reference for the development of 2 μm wavelength band integrated photonics.

## 1. Introduction

With the emergence of devices such as hollow-core photonic bandgap fibers [1,2] and thulium-doped fiber amplifiers [3,4] operating at 2 μm, the 2 μm wavelength band has gradually developed the potential to become a new communication band. Moreover, with the rapid development of noninvasive glucose measurement technology [5,6] and biosensing technology [7,8], there is an increasing demand for high-performance 2 μm photonic devices [9,10,11,12,13]. Furthermore, integration of those single devices on the same platform is attractive due to the motivation of realizing a compact module/system for commercialization. Meanwhile, a silicon on insulator (SOI) platform with high integration, low loss and compatibility with the CMOS process has been widely used in photonic integrated circuits [14,15,16,17,18].

In this platform, the mode field size of a single-mode Si waveguide is at the submicron level, and the mode field size of a single-mode optical fiber is at the micron level; direct coupling between the fiber and waveguide leads to high loss. Spot size converters (SSCs) and grating couplers are two devices used for fiber to Si waveguide coupling. Grating couplers are preferred for a wafer-level test due to their flexible test positioning and low alignment accuracy satisfying the requirement of commercial applications [19]. SSCs are not sensitive to polarization and demonstrating lower coupling losses, which is more suitable for the non-polarization-sensitive and low loss application scenarios such as polarization multiplexing and quantum key distribution [20,21]. Meanwhile, with the application of a V-groove structure on the chip to fix the position and spacing of optical fibers, the difficulty of alignment between the SSC and the optical fiber has been reduced [22]. Therefore, it is meaningful to focus on the study of SSCs for the realistic application of a silicon photonic chip coupled with an optical fiber. In addition, SSCs have a wide range of applications in chip–chip coupling and the coupling of waveguides of different sizes.

Typically, SSC structures include tapered waveguides [23,24,25], tridents [26,27,28], subwavelength gratings [22,29] and other structures [30,31,32,33]. In 2007, Kazuo Shiraishi et al. proposed an SSC that integrates a vertical tapered waveguide with a horizontal tapered waveguide [24]. It can extend the spot size in a Si waveguide from 0.54 × 0.38 μm^2^ to 5.1 × 9.2 μm^2^ and improve the coupling efficiency with an optical fiber; it is mainly applied for thick silicon platforms where the waveguide thickness is on the level of microns. However, more applications are based on 220 nm thick SOI platforms due to the 220 nm thick SOI platform demonstrating excellent single-mode characteristics and ease of integration with passive photonic devices. An invertly tapered structure is needed to achieve a lossless mode conversion before the optical fiber and the Si single-mode waveguide. In 2011, Qing Fang et al. designed a 1.55 μm wavelength band low-loss SSC based on a cantilevered structure, which utilizes a SiO_2_ waveguide to initially compress the spot size before coupling it into a tapered waveguide, with a coupling loss as low as 1 dB for coupling to a single-mode fiber (SMF) with a mode field diameter (MFD) of 10.5 μm [31]. In 2021, An He, Xuhan Guo et al. utilized a three subwavelength grating strip waveguide to increase the mode field matching between an SSC and an SMF (MFD: 10 μm), with a minimum coupling loss of 1.44 dB in the C-band [29]. All of the above devices are 1.55 μm wavelength band SSCs. Updates for 2 μm wavelength band SSCs are as follows. In 2021, Xi Wang, Jiangbing Du et al. proposed a 2 μm SSC for the input and output of a 2 μm Mach–Zehnder modulator with a loss of 2.5 dB [34]. In 2022, Sumei Xu, Qize Zhong et al. proposed a 2 μm SSC based on AlN waveguides with an idealized 90° sidewall angle, and the loss of the device was 1.54 dB obtained by simulation [35]. However, it is challengeable to realize an AlN waveguide with a 90° sidewall angle. In addition, the trident structure has not been applied to the 2 μm wavelength band. In summary, the SSCs at 2 μm wavelengths still demonstrate either a relatively high coupling loss or a complex process for fabrications. Improvements need to be achieved for such components to realize high coupling efficiency while satisfying the process standard of the commercial fab.

In this paper, a 2 μm wavelength band SSC based on a special trident structure is proposed and fabricated. It is coupled to a lensed fiber with an MFD of 5 μm, and the coupling loss of the SSC is 0.8 dB at a 2 μm wavelength simulated by FDTD. The lowest experimental result of the SSC loss is 1.425 dB at 2 μm, which is close to the simulation result. Furthermore, the minimum waveguide width of the device is set as 220 nm, which is suitable for the lithographic resolution of current commercial silicon photonics fabs, and it has the possibility of large-scale fabrication.

## 2. SSC Design and Simulation

The structure of the 2 μm SSC is shown in Figure 1. The first segment of the tapered waveguide in the trident structure has a right-angled trapezoidal cross-section, which can reduce the loss of the device. The SSC enables efficient coupling of the optical fiber to the single-mode Si waveguide. The thickness of the Si waveguide is 220 nm, the SiO_2_ thickness for both the BOX and cladding layer is 2 μm, and the size of the single-mode Si waveguide is 500 nm (width) × 220 nm (height). The device is simulated by the FDTD and EME modules of Lumerical software (version number: 2020 R2.4).

The SSC is mainly used for transmitting TE polarized light because the single-mode waveguide has a weaker ability to confine 2 μm TM polarized light results in very low transmission efficiency. Figure 2 shows the transmission efficiency of different width waveguides for both polarizations. It can be found that the single-mode waveguide with a width of 0.5 μm can only confine 65% of TM polarized light, which leads to a high loss of TM (~1.8 dB). Therefore, this SSC is more useful for transmitting TE polarized light, and in experiments, the polarization of the input light can be determined by coupling losses (with the lowest loss occurring for TE polarization).

At present, the lensed fiber has been used in the 2 μm band operation, and the performance of low coupling loss has been proved by experiments [36]. The lensed fiber is also utilized in this work. The structure and parameters of the lensed fiber provided by the vender (Corning/SMF-28E+LL/9/125/250/100KPSI-CX000, CXFIBER, Wuhan, China) is shown in Figure 3a,b [37]. The spot size of the lensed fiber is 3 μm at 1.55 μm, and the spot size of a lensed fiber at 2 μm is not provided. Therefore, the spot size of the lensed fiber at the wavelength of 2 μm needs to be obtained by simulation. The refractive indices of the fiber core can be obtained by the Sellmeier equation [38]:(1)$$n^{2}\left(\lambda \right)=1+\sum_{i=1}^{3}\frac{a_{i}\lambda^{2}}{\lambda^{2}-b_{i}^{2}}$$
where *n* is the refractive index, *λ* is the wavelength, and *a_i_* and *b_i_* are the Sellmeier coefficients, and the values of *a*_1_, *a*_2_, *a*_3_, *b*_1_, *b*_2_ and *b*_3_ in Corning’s single-mode fiber are shown in Table 1. The refractive indices of the fiber cladding can be obtained by the *NA* formula:(2)$$\mathrm{NA}=\sqrt{n_{\mathrm{co}}^{2}-n_{\mathrm{cl}}^{2}}$$
where *NA* is the numerical aperture (the *NA* of Corning’s single-mode fibers is typically 0.13), *n_co_* is the core refractive index and *n_cl_* is the cladding refractive index. From Equations (1) and (2), the refractive indices of the core and cladding are set as 1.4447 and 1.4388 at 2 μm wavelengths, respectively. Figure 3c shows the lensed fiber MFD simulated by the FDTD solution, which is around 5 μm at 2 μm and it is approaching a type of Gaussian distribution. Therefore, the incident SSC diameter of the SSC is set as 5 μm in the following design and simulation process. Additionally, the incident spot diameter of 8 μm is applied for the loss of the SSC coupling with the single-mode fiber as the reference.

The loss of the whole device consists of two parts. The first part is the mode field mismatch loss of the lensed fiber and the trident structure, and the second part is the transmission loss of the waveguide in the trident. In theory, the mode field matching efficiency can be calculated from the overlap integral of the two mode fields. The mode overlap integral can be calculated by Equation (3) [39]:(3)$$\eta =\frac{|\int {E_{1}E_{2}\mathrm{dA}|}^{2}}{|\int {E_{1}|}^{2}\mathrm{dA}|\int {E_{2}|}^{2}\mathrm{dA}}$$
where *E_1_* and *E*_2_ are the Gaussian mode field distribution at the output end of the lensed fiber (Figure 4a) and the mode field distribution of the TE fundamental mode at the start of the waveguide section of the trident structure (Figure 4b), respectively.

In the device, the width of the tapered waveguide tip (W_tip_) and the gap between the tapered waveguide tips (W_gap_) can affect the mode field distribution at the receiving end of the SSC; the mode field matching between the lensed fiber and the SSC can be improved by optimizing these two parameters. Here, the coupling loss is used to characterize the fiber and SSC mode field matching results. Figure 5a,b show the variation in device loss with W_tip_ and W_gap_, respectively (when one parameter is varied and the other parameters are fixed at optimal values). When W_tip_ is fixed as 0.22 μm and W_gap_ is 1.5 μm, the coupling loss of the SSC could reach the minimum.

L_1_ and L_2_ affect the transmission loss of the tapered waveguide. The width of the tapered waveguide on both sides is extended from 0.22 μm to 0.4 μm, and, the lengths for these multiple tapered waveguides are optimized to achieve the minimum coupling loss of the whole device. Figure 5c shows the variation in device loss with the length of the tapered waveguide on both sides simulated by the FDTD solution, and the device demonstrates the lowest loss when L_1_ is 80 μm. The central waveguide width is tapered from 0.22 μm to 0.5 μm for adiabatic mode contraction for coupling to the single-mode SOI waveguide. Figure 5d shows the relationship between the transmission loss of the tapered waveguide and L_2_ simulated by the EME solution, and the tapered waveguide satisfies the adiabatic transmission condition when the waveguide length is more than 200 μm. Table 2 shows the final values for each parameter.

Figure 6 shows the simulation results of the SSC, demonstrating that the light is efficiently transmitted into the Si waveguide through the trident structure. When the MFD is 5 μm for the single-mode fiber coupling, the coupling loss of the SSC can be less than 0.9 dB in the wavelength range of 1.95~2.05 μm, and the coupling loss is 0.8 dB at 2 μm. When the MFD is 8 μm for the single-mode fiber coupling, the coupling losses can be confined within 1~1.1 dB at the same wavelength region, and the coupling loss is 1.07 dB at 2 μm. Finally, the transmittance results of the designed SSC based on a special trident structure (the first segment of the tapered waveguide has a right-angled trapezoidal cross-section) and the SSC based on a general trident structure (the structure is designed by replacing the first segment of the tapered waveguide in the special trident structure to a tapered waveguide with an isosceles trapezoidal cross-section) are compared. The transmittance results of the two structures are shown in Figure 7, and it can be found that our designed structure achieves improvement.

## 3. Experiment and Analysis

### 3.1. Experiment Setup and Characterization Results

The SSC was fabricated by the CUMEC (Chongqing United Microelectronics Center, Chongqing, China) on the CSiP180Al technology platform with the standard process of deep UV lithography [40]. The test setup is shown in Figure 8, and the laser was a single-mode laser diode (LDs, EP2000-DM-B by Eblana Photonics at Dublin, Ireland). The polarization controller was applied to adjust the polarization of the input light. The optical power detection in the system was achieved by an optical spectrum analyzer (OSA, AQ6375 by Yokogawa Electric Corporation at Tokyo, Japan). The transmission fiber and the lensed fiber for coupling were fibers operating in the C-band. The alignment of the lensed fiber and the SSC was achieved by means of a dual-axis motorized translation stage (MT1-Z9 by Thorlabs, Newton, NJ, USA), which has a minimum step size of 30 nm to satisfy the requirements of the test.

Since the optical fibers used for transmission and coupling in the system are operating in the C-band, it may cause high losses by transmitting light of the 2 μm wavelength band. In order to obtain an accurate loss of the SSC, the impact of these factors on the results needs to be considered. In this system, the optical power of the PC output port is set to the input optical power (P_in_), and the output port of the second segment of the lensed fiber is set to the output optical power (P_out_), so the transmission loss of the lensed fiber is a key consideration. Due to the minimum loss being the focus of this paper, maximum alignment accuracy was achieved by observing the best transmission efficiency during the experiment, which was adopted in both fiber-to-fiber coupling and total loss characterizations. Therefore, an experiment about the fiber-to-fiber coupling (excluding the device) was conducted before the total loss measurement (involving the device) to figure out the two fibers’ transmission losses. This is a reasonable method that has been published in Reference [41]. The alignment of the two lensed fibers with the same specifications in the vertical and horizontal directions was achieved using a motorized translation stage, as shown in Figure 9. Then, by adjusting the distance between the two lensed fibers, we could obtain a minimum loss.

Set (P_in_ − P_out_)/2 as the total loss including the transmission loss of the lensed fiber, the loss of the SSC and the loss of the straight waveguide used to connect the two SSCs. The total loss of the device and the loss of each discrete component are shown in Table 3. The total loss is 2.55 dB/facet, the transmission loss of the lensed fiber is 0.75 dB/facet and the loss of the straight waveguide is 0.375 dB (considering the waveguide length of 0.25 cm and transmission loss of 1.5 dB/cm provided by CUMEC [40]). Therefore, the coupling loss of the SSC is estimated as 1.425 dB, which is approaching the simulation result of 0.8 dB.

The comparison between our work and the currently published results is also listed in Table 4. It can be seen that our designed SSC achieves the lowest coupling loss between the fiber and the Si single-mode waveguide.

### 3.2. Analysis of the Difference between the Simulation and Experiment Results

It can be seen that there is about 0.6 dB difference between the measurement and simulation results. There are two possible reasons for this result. One is the transmission loss of the tapered waveguide in the device, which can be estimated to be 0.054 dB considering the transmission loss of the straight waveguide (1.5 dB/cm). This loss is negligible compared with the total loss difference. The second is that there is some difference between the original design and final fabricated device (See Figure 10a). In the actual fabrication process, a 17 μm straight waveguide is added to the front section of the SSC to suppress the effect of edge processing during chip fabrication. This may affect the coupling loss of the fabricated device. Figure 10b shows the simulated results of the coupling loss of the two structures demonstrating a difference of ~0.47 dB between the original design (0.8 dB) and the final fabricated device (1.27 dB) at 2 μm wavelengths. Therefore, it can be found that the main reason for the loss difference is the pattern change during the device’s fabrication.

## 4. Conclusions

In this paper, we design and fabricate a low-loss 2 μm SSC by optimizing the cross-section of the first segment of the tapered waveguide structure in the trident structure. Simulations demonstrate a coupling loss of 0.8 dB at a wavelength of 2 μm when the light beam has an MFD of 5 μm, and when the light beam has an MFD of 8 μm, the coupling loss is 1.07 dB at 2 μm. According to the experimental results, the SSC has the lowest loss of 1.425 dB, which is close to the simulation result. Furthermore, through the feedback from the vendor of the device, a SiO_2_ cladding with a width of 2–3μm has been added before the silicon waveguide, and the edge treatment of this region will not directly affect the trident structure. Therefore, further work is considered to remove the straight waveguide in the front of the trident. Overall, the designed device shows great potential in fiber-to-chip coupling, with the minimum waveguide width of the device set as 220 nm, which is suitable for the lithographic resolution of current commercial silicon photonics fabs, and has the possibility of large-scale fabrication.

## Figures and Tables

**Figure 1 micromachines-15-00530-f001:**
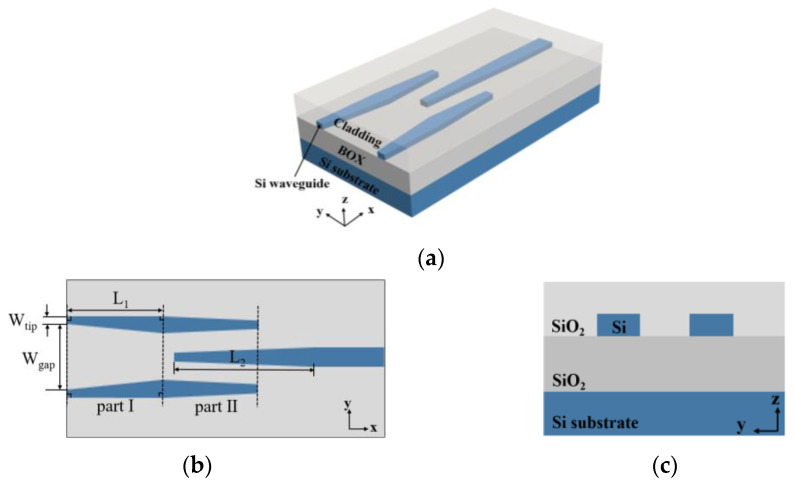
The structure of the SSC: (**a**) 3D view; (**b**) top view; (**c**) side view.

**Figure 2 micromachines-15-00530-f002:**
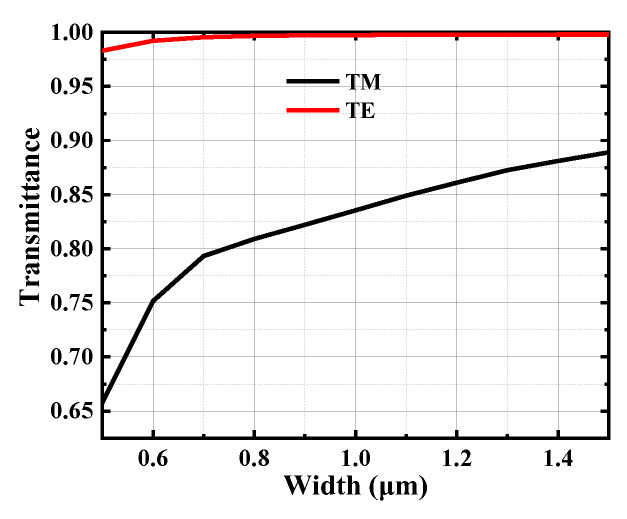
The transmission efficiency of different width waveguides for TE and TM.

**Figure 3 micromachines-15-00530-f003:**
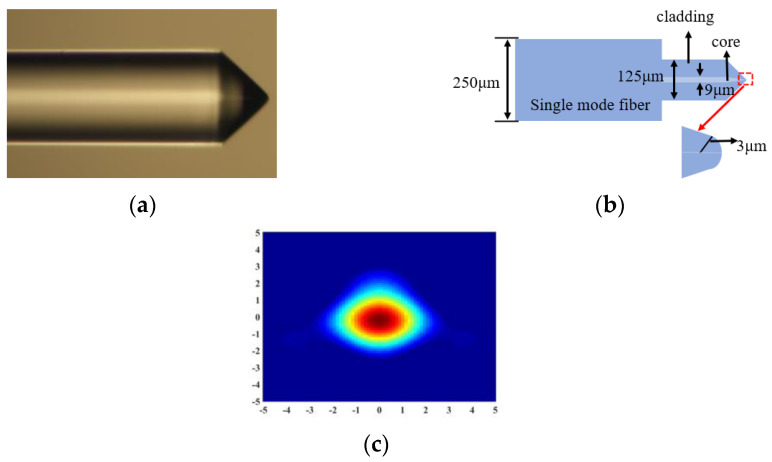
Structure and simulation results of lensed fiber. (**a**) Lensed fiber structure under the microscopic observation, (**b**) Lensed fiber structure diagram, (**c**) spot size at 2000 nm.

**Figure 4 micromachines-15-00530-f004:**
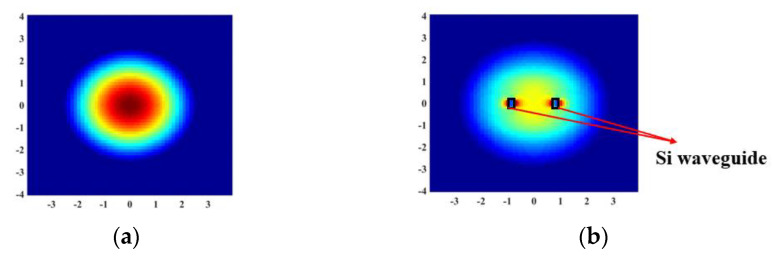
The field distributions of different modes. (**a**) The Gaussian mode field distribution at the output end of the lensed fiber. (**b**) The mode field distribution of the TE fundamental mode at the start of the waveguide section of the trident structure.

**Figure 5 micromachines-15-00530-f005:**
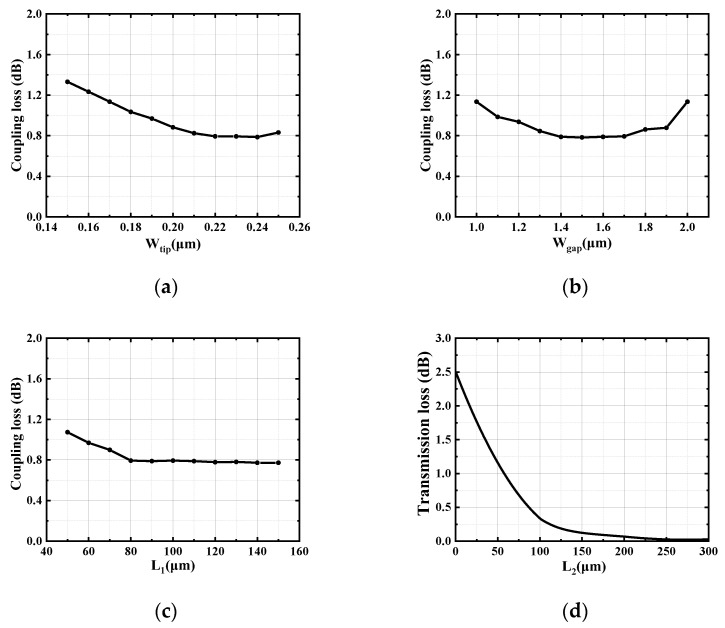
The effects of each parameter on the performance of the SSC when one parameter is varied and the other parameters are fixed at optimal value. (**a**) W_tip_, (**b**) W_gap_, (**c**) L_1_, (**d**) relationship between transmission loss of tapered waveguide and L_2_.

**Figure 6 micromachines-15-00530-f006:**
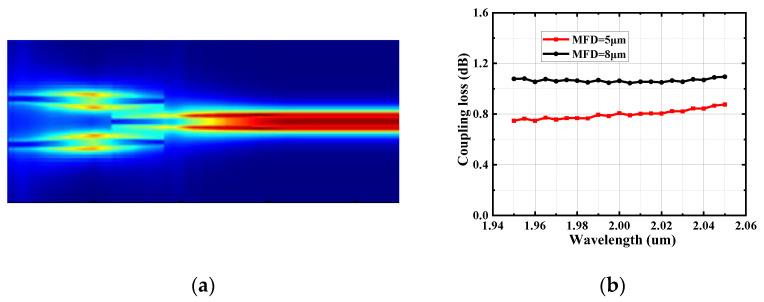
Simulated results for the SSC around 2 μm wavelengths (TE mode). (**a**) The electric field distribution for the light passing through the device. (**b**) The simulated result for the coupling loss (red line: MFD = 5 μm, black line: MFD = 8 μm).

**Figure 7 micromachines-15-00530-f007:**
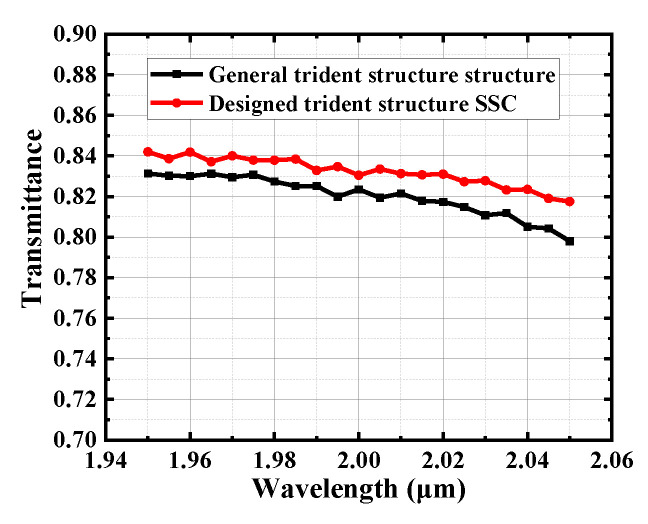
Comparison of transmittance results for our designed trident structure SSC and general trident structure SSC.

**Figure 8 micromachines-15-00530-f008:**
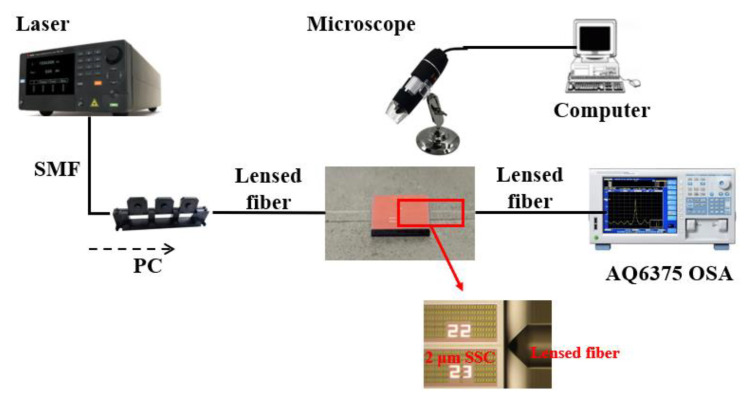
Experiment setup.

**Figure 9 micromachines-15-00530-f009:**
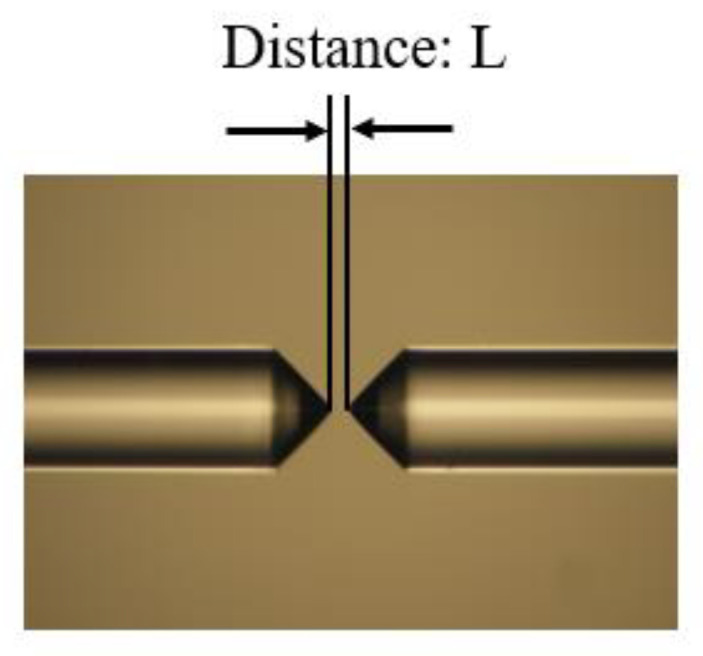
The schematic of the alignment of two lensed fibers in the vertical and horizontal directions.

**Figure 10 micromachines-15-00530-f010:**
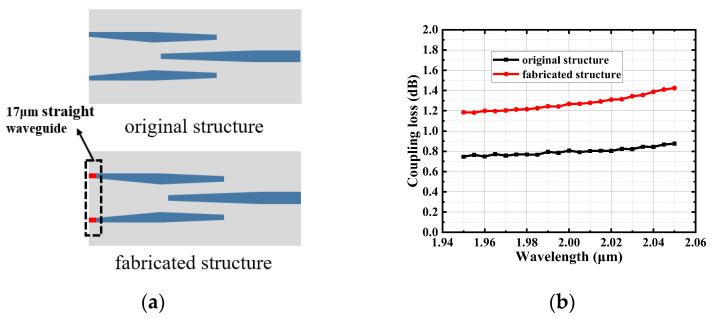
The difference between the original design and fabricated device due to the fabrication error. (**a**) Difference in the device structure before and after fabrication. (**b**) Simulated results for the coupling loss of the device before and after fabrication.

**Table 1 micromachines-15-00530-t001:** The values of Sellmeier coefficients [38].

Parameter	Value
*a* _1_	0.49211
*a* _2_	0.62925
*a* _3_	0.59202
*b* _1_	0.04807
*b* _2_	0.11275
*b* _3_	8.29299

**Table 2 micromachines-15-00530-t002:** Final values for each parameter.

Parameter	Value
W_tip_	0.22 μm
W_gap_	1.5 μm
L_1_	80 μm
L_2_	200 μm

**Table 3 micromachines-15-00530-t003:** Loss of the device.

	Loss (min)
Total loss/facet	2.55 dB
Loss of the coupling lensed fiber/end	0.75 dB
Loss of the straight waveguide (1.5 dB/cm [40])/facet	0.375 dB
Loss of the device/facet	1.425 dB

**Table 4 micromachines-15-00530-t004:** Comparison of coupling loss for different 2μm wavelength band SSCs.

Reference	MFD (μm)	Coupling Loss	Simulation or Experimental
[34]	8	2.5 dB	Experimental
[35]	8	1.54 dB	Simulation
This work	8/5	1.1 dB/1.425 dB	Simulation/Experimental

## Data Availability

The original contributions presented in the study are included in the article, further inquiries can be directed to the corresponding author.

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
