# Peer review of "A 2 μm Wavelength Band Low-Loss Spot Size Converter Based on Trident Structure on the SOI Platform"

_micromachines, 2024, doi:10.3390/mi15040530_

Round 1
Reviewer 1 Report
Comments and Suggestions for Authors
The manuscript presents a 2μm spot size converter based on a trident structure, achieving an insertion loss of 1.425dB/facet. Spot size converters play a crucial role in chip packaging with fibers. While the work is interesting, certain descriptions lack clarity. Below are my detailed comments that need to be addressed.
1. Has the trident structure been previously demonstrated in similar materials or wavelength bands? What differentiates this work from existing research in the field?
2. The authors designed a trident structure SSC for a fiber with MFD of 8μm. Why not demonstrate it experimentally?
3. Considering that flat facet fibers are more commonly used in commercial products and have less variability in MFD compared to lensed fibers, it would be prudent to discuss the choice of lensed fibers in this context.
4. In Figure 1b, the trapezoid does not appear to have a right-angle shape, which might require clarification or adjustment for accuracy.
5. Why was the width of the tapered waveguide extended to 0.4μm? Is this value optimal, and why wasn't it explored as a variable?
6. There seems to be a discrepancy in the calculation of SSC coupling loss. The total loss measurement includes two fiber transmission losses, two fiber-to-SSC coupling losses, and waveguide loss. However, it seems that the fiber-to-fiber coupling loss, which does not occur in the DUT measurement and is sensitive to alignment, is not accounted for. This might lead to underestimation of the fiber-to-SSC coupling loss in actual operation. Clarification on this discrepancy would be helpful.
Reviewer 2 Report
Comments and Suggestions for Authors
In this paper, an optimized trident structure has been applied to the SOI platform to work as the spot size converter at 2 um wavelength. Simulation and experimental results demonstrate lower loss than the recently published results. While, some question needs to be answered before the finial acceptance.
Q1: In the experiment, how could you figure out the polarization state of the input light?
Q2: Is there any potential method for reducing the loss from the extra waveguide before the designed trident structure?
Reviewer 3 Report
Comments and Suggestions for Authors
I have the following remarks and suggestions:
1) I am not sure why author didn't consider tapered waveguide to directly couple the light from fiber to the waveguide. I mean to say, keep the input section of the waveguide to 1-2 microns and coupling the 5 microns MFD to the waveguide. And then eventually tapered down the waveguide width to the single mode dimensions i.e., 400 to 500 nm. I suggest the author add valid reasoning on this point in the paper to suggest the need of this work.
2) The photograph of the fiber coupling to the device should be shown in zoomed dimensions. I mean to say, show the readers where the tip of the tapered fiber was placed. In between the two waveguides or coupling to one waveguide?
3) There is no information on the fabrication process, which techniques are used? Mention in detail.
4) The SEM images of the fabricated device should be added.
5) The potential applications of the proposed mode size converter should be added in the Introduction section.
6) The author said "Spot size converters 32 (SSCs) and grating couplers are two devices used for fiber to Si-waveguide coupling. 33 Comparing with the grating couplers, SSCs are more suitable for commercial production 34 owing to their low coupling loss, polarization insensitivity [19,20], and easy packaging 35 [21,22]." However, for commercial applications grating couplers are preferred due to the less complication in fiber-to-waveguide alignment. Therefore, I suggest the author should provide a valid reasoning for this claim.
7) In Figure 1, the XYZ axis label should be added.
Comments on the Quality of English Languagenone.
Round 2
Reviewer 1 Report
Comments and Suggestions for Authors
The comments are barely addressed. The manuscript is OK to be published.
Reviewer 3 Report
Comments and Suggestions for Authors
I am willing to accept the paper in its current form.
Comments on the Quality of English Languagenone.